# Targeting Prohibitins to Inhibit Melanoma Growth and Overcome Resistance to Targeted Therapies

**DOI:** 10.3390/cells12141855

**Published:** 2023-07-14

**Authors:** Ahmad Najem, Mohammad Krayem, Serena Sabbah, Matilde Pesetti, Fabrice Journe, Ahmad Awada, Laurent Désaubry, Ghanem E. Ghanem

**Affiliations:** 1Laboratory of Clinical and Experimental Oncology (LOCE), Institut Jules Bordet, Université Libre de Bruxelles, 1000 Brussels, Belgium; mohammad.krayem@bordet.be (M.K.); serena.sabbah@hubruxelles.be (S.S.); matilde.pesetti@student.unisi.it (M.P.); fabrice.journe@bordet.be (F.J.); ahmad.awada@bordet.be (A.A.); ghanem.elias.ghanem@ulb.be (G.E.G.); 2Department of Medical Oncology, Institut Jules Bordet, Université Libre de Bruxelles, 1070 Brussels, Belgium; 3Center of Research in Biomedicine of Strasbourg, Regenerative Nanomedicine (UMR 1260), INSERM, University of Strasbourg, 67000 Strasbourg, France; desaubry@unistra.fr

**Keywords:** melanoma, prohibitins, mitochondria-targeted agents (MTA), drug resistance, MAPKi, therapeutic strategy

## Abstract

Despite important advances in the treatment of metastatic melanoma with the development of MAPK-targeted agents and immune checkpoint inhibitors, the majority of patients either do not respond to therapies or develop acquired resistance. Furthermore, there is no effective targeted therapy currently available for BRAF wild-type melanomas (approximately 50% of cutaneous melanoma). Thus, there is a compelling need for new efficient targeted therapies. Prohibitins (PHBs) are overexpressed in several types of cancers and implicated in the regulation of signaling networks that promote cell invasion and resistance to cell apoptosis. Herein, we show that PHBs are highly expressed in melanoma and are associated with not only poor survival but also with resistance to BRAFi/MEKi. We designed and identified novel specific PHB inhibitors that can inhibit melanoma cell growth in 3D spheroid models and a large panel of representative cell lines with different molecular subtypes, including those with intrinsic and acquired resistance to MAPKi, by significantly moderating both MAPK (CRAF-ERK axis) and PI3K/AKT pathways, and inducing apoptosis through the mitochondrial pathway and up-regulation of p53. In addition, autophagy inhibition enhances the antitumor efficacy of these PHB ligands. More important, these ligands can act in synergy with MAPKi to more efficiently inhibit cell growth and overcome drug resistance in both BRAF wild-type and mutant melanoma. In conclusion, targeting PHBs represents a very promising therapeutic strategy in melanoma, regardless of mutational status.

## 1. Introduction

Despite the clinical success and the marked initial responses with immune checkpoint blockade and the combination of BRAF and MEK inhibitors, there are no durable responses, and the development of acquired resistance inhibitors is inevitable [1,2]. Furthermore, there is no effective targeted therapy currently available for BRAF wild-type melanoma, accounting for approximately 50% of cutaneous melanoma. Hence, the identification of key therapeutic targets is very crucial to improve the targeted therapies of metastatic melanoma.

Prohibitins (PHBs) are scaffold proteins that are overexpressed in several tumor types [3,4]. The high expression of PHBs is significantly correlated with tumor metastasis and poor prognosis in neuroblastoma, glioblastoma, lung, pancreatic, bladder, prostate, and breast cancers [5,6,7,8,9,10,11]. The latter plays a crucial role in cancer progression by modulating many signaling pathways involved in cell survival and cell invasion [5,6,7,8,9,10,11]. PHBs have an essential role in the RAS-driven activation of the MAPK/ERK1/2 pathway via CRAF [12], the key mediator in wild-type BRAF melanoma where there are no effective therapies. Moreover, PHBs can also activate the PI3K/AKT pathway [9,12], a main downstream signaling network in NRAS mutant cells. This important pathway is also involved in the intrinsic and acquired resistance to MAPKi in BRAF mutant melanoma [13]. Moreover, PHBs in the mitochondria have an anti-apoptotic role and their accumulation can lead to chemoresistance [14]. Furthermore, PHBs are implicated in tumor invasion and metastasis. Indeed, PHBs can promote cell invasion via ROCK2 and RACK1 and their associated signaling pathways [9,15,16]. Also, PHBs enhance EMT, stemness, and therefore, the undifferentiated/invasive phenotype in several tumor types [7,8,9,15,17]. In addition, PHBs are key regulators of HES1, which has an important function in cancer metastasis, and chemotherapy resistance by promoting EMT [18,19]. All of the above suggest that PHBs represent a very attractive therapeutic target that deserves to be evaluated in melanoma. Several PHB inhibitors have been identified [20]. The natural compounds called flavaglines and their derivatives display potent anticancer activities, can lead to the diminution of the amount of mitochondria-associated PHBs and can interfere with the membrane localization of PHB. Of note, the latter did not show any sign of toxicity in mice studies and healthy cells [21,22]. Fluorizoline is another PHB ligand that induces apoptosis in several cancer cells. Accordingly, it was shown that Fluorizoline promotes p21 expression and thereby inhibits cancer growth [23]. JI130 is another PHB ligand that promotes the interaction between the transcription factor HES1 and PHB2 in the cytosol to promote G2/M cell cycle arrest. Importantly, JI130 reduced by half the tumor growth in a pancreatic tumor xenograft model. Whether this class of compounds modulates the interaction between PHB2 and some of its other partners remains an open question [24].

In our previous screening study, we showed that targeting PHBs using novel specific PHB inhibitors (melanogenin derivatives including MEL9, MEL41, and MEL56) promotes melanogenesis and apoptosis in melanoma cells. These PHB ligands termed melanogenin analogs are highly active, specific, non-toxic in normal cells, and are more drug-like analogs of melanogenin. Indeed, we found that among 57 new melanogenin analogs screened, these candidates can regulate cell differentiation via LC3-II and induce cell death in a panel of human cancer lines including melanoma. Also, we found that these melanogenin derivatives can bind to PHBs and downregulate the cellular levels of PHB protein [25].

Herein, we provide evidence of the potential role of PHBs as new targets in melanoma and the rationale to target PHBs in combination with MAPK inhibitors as a novel promising therapeutic strategy within different genomic subtypes.

## 2. Materials and Methods

### 2.1. Patients and Tissue Collection

Skin and lymph node metastases (*n* = 37, male/female = 15/22) were collected from patients with stage III or IV melanoma who underwent surgery at Institut Jules Bordet (Brussels, Belgium) between 1998 and 2009, ensuring a sufficient follow-up of the patients. Of note, at the time of sample collection, patients were not treated by any targeted therapies or immunotherapies, while chemotherapy was associated with very modest clinical responses, thus avoiding any effects of treatment on patient survival. Surgical samples (mean size 10 mm, no necrosis) were randomly collected with no inclusion or exclusion criteria. The metastatic tissue samples were snap-frozen in liquid nitrogen and stored at −80 °C and were dedicated to RNA extraction and PCR. This study was approved by the ethics committee of Institut Jules Bordet (CE1959) and performed in accordance with the REMARK guidelines. The samples were registered to the Biobank of Institut Jules Bordet (BB190035). The median age of the patients at diagnosis of primary melanoma was 57.4 years old (range 25.8–87.2). The median duration for progression-free survival (PFS) was 1.2 years (range 0.1–25.7), and 4.3 years (range 0.8–28.6) for overall survival (OS).

### 2.2. RNA Extraction and Real-Time PCR of Patient Samples

Frozen samples were homogenized using the FastPrep-24 homogeniser system with lysing matrix D (MP Biomedicals, Illkirch Cedex, France) in RLT buffer supplemented with β-mercaptoethanol (RNeasy Mini Kit, Qiagen, Venlo, The Netherlands) at 4 °C. Centrifugation with RNeasy spin column separated melanin from the total RNA. After the washing steps, RNA was collected in RNase-free water and RNA concentrations were evaluated using a NanoDropTM 1000 spectrophotometer (Thermo Scientific, Waltham, MA, USA). The RNA quality of each sample was assessed based on the RNA profile generated by the bioanalyzer (Agilent Technologies, Santa Clara, CA, USA). The PHB1 and PHB2 mRNA expression was quantified by real-time PCR. Complementary DNA was synthesized using a standard reverse transcription method (qScript cDNA SuperMix, Quanta Biosciences, Gaithersburg, MD, USA). Real-time PCR reactions were performed using the SYBR Green PCR Master Mix (Thermo Scientific, USA) and sequence-specific primer sets for PHB1 and PHB2 (Thermo Scientific). The amplification was performed using QuantStudi^TM^ 3 Thermo Fisher Scientific Real-Time PCR system by performing 40 cycles. The messenger RNA expression of PHBs was normalized to 18S (loading control) and presented using the 2-∆CT method.

### 2.3. Effectors

Dabrafenib, Trametinib (AS-703026), Sunitinib, and chloroquine (CQ) were from Selleck Chemicals. They were dissolved, according to the manufacturer’s recommendations, aliquoted, and stored at −20 °C. PHB ligands were synthesized and obtained from the group of Laurent Désaubry (University of Strasbourg) and dissolved in DMSO.

### 2.4. Melanoma Cell Culture

Human melanoma cell lines used in this study were all established in the Laboratory of Clinical and Experimental Oncology (LOCE-Institut Jules Bordet). Cells were cultured in Ham’s F10 medium supplemented with 10% heat-inactivated fetal calf serum (FCS), and with L-glutamine, penicillin, and streptomycin at standard concentrations (Thermo Fisher Scientific, Waltham, MA, USA) in humidified air with 5% CO_2_ at 37 °C [26]. The cell culture medium was renewed every 2–3 days. Once the cells were at or near confluence, they were subcultured. Melanoma cells were regularly checked for mycoplasma contamination using MycoAlert^®^ Mycoplasma Detection Kit (Lonza, Basel, Switzerland). Cell Line Authentication was performed using STR profiling with AmpFLSTR^TM^ Identifiler^TM^ PCR Amplification Kit (Thermo Fisher Scientific). DNA isolation was carried out from a cell pellet of 1 × 10^6^ cells and 16 independent PCR systems were investigated and analyzed (Eurofins Genomics, Ebersberg, Germany).

### 2.5. Proliferation Assay

Cell proliferation was assessed by crystal violet assay [27]. All cells were seeded in 96-well plates (8 × 10^3^ cells/well). One day after plating, the culture medium was replaced by a fresh one either containing effectors or not, depending on the experimental conditions, and cells were further cultured for 3 days.

### 2.6. Annexin V Assay/Apoptosis Determination

Apoptotic cells were measured using Annexin V-FITC Apoptosis Detection Kit (Miltenyi Biotec, Gaithersburg, MD, USA), according to the manufacturer’s recommendations. Briefly, cells were seeded in six-well plates and allowed to adhere for 24 h. After plating, the culture medium was replaced by a fresh one either containing effectors or not, and cells were further incubated for 2 days before assay. For the detection of apoptosis, cells were collected, centrifuged, washed, and resuspended in 100 µL 1 × Binding Buffer (Miltenyi Biotec). After the addition of 5 µL annexin-V-FITC, cells were incubated for 20 min and then analyzed by flow cytometry (FACS Beckman Coulter Navios, Brea, CA, USA).

### 2.7. Measurement of Mitochondrial Membrane Potential

The mitochondrial membrane potential (Δψm) was assessed using MitoProbe^TM^ DiIC1 according to the manufacturer’s recommendations. For this purpose, cells were harvested 2 days after treatment and incubated with 5 μL of 10 μM DiIC1 at 37 °C for 30 min and then analyzed by flow cytometry (Beckman Coulter Navios).

### 2.8. Caspase 3/7 Activity Assay

Caspase activity was measured using the CellEvent^TM^ Caspase-3/7 green assay kit (Thermo Fisher Scientific, USA) according to the manufacturer’s recommendations. Briefly, cells were harvested, centrifuged, and washed 2 days following the treatment. Then, 1 μL of CellEventTM Caspase-3/7 green detection reagent was added to all samples that were incubated for 30 min and then analyzed by flow cytometry (Beckman Coulter Navios).

### 2.9. Evaluation of 3D Melanoma Spheroid Size and Viability

Spheroids were generated in 96-well Ultra-Low-Attachment round-bottom plates (Corning^®^ ULA plate, Corning, NY, USA). HBL melanoma cells were seeded at concentrations of 1000, 2000, and 4000 cells/well, and spheroid growth was monitored for 3 days to evaluate the effect of treatment. The growth of the spheroids was investigated using an inverted microscope (Nikon diaphot inverted microscope). The spheroids’ size (surface area) was assessed using ImageJ software v6. The cell viability was determined using an MTT assay (Sigma Aldrich, St. Louis, MO, USA). The latter was carried out after 72 h, MTT solvent was added to each well, and plates were incubated at 37 °C with CO_2_ at 5% and a humidified atmosphere for 4 h. Then, DMSO was added and the optical density was determined through a spectrophotometric microplate reader (Thermo Scientific Multiskan EX).

### 2.10. Western Blot Analysis

Cells were plated in Petri dishes (3 × 10^6^ cells/dish) in culture medium. One day after plating, the culture medium was replaced by a fresh one and further left for 2 days. Then, cells were either exposed or not to effectors for 24 h. Cells were lysed using a detergent cocktail (Thermo Fisher Scientific) and extracted proteins were analyzed by Western blot (28). Immunodetections were performed using antibodies raised against PHB1, PHB2 (E1Z5A), HES1 (D6P2U), pCRAF (Ser338) (56A6), c-Raf (D4B3J), pAKT (Ser473), AKT, p21 (12D1), LC3B (D11), AXL (C89E7), ZEB1 (D80D3), and MMP9 (all from Cell Signaling Technology), in addition to p53 (DO-1), ERK (Tyr 204) (E-4), ERK2(C-14), β-actin (MAB1501R) (1/5000) (Merck) (details on electrophoresis and immunodetection have been described previously) [13]. Stained band intensities were analyzed and compared using Image J.

### 2.11. Autophagy Detection

Autophagy vacuoles were assessed using the Autophagy Assay Kit (ab139484, Abcam, Cambridge, UK) according to the manufacturer’s protocol. Briefly, after 24 or 48 h of treatment, cells were harvested and washed with PBS. Then, the green detection reagent from the kit was added to all samples that were incubated for 30 min and then analyzed by flow cytometry (Beckman Coulter Navios). The increase in the green fluorescence signals is represented by the shift in the fluorescence peak along the abscissa axis. Data are represented as the mean of fluorescence intensity ± SEM of three independent experiments.

### 2.12. Cell Migration Assay

Cell migration was assessed using transwell inserts (Corning, USA). Briefly, a total of 1 × 10^4^ cells in a serum-free culture medium were seeded into the upper chamber of a transwell filter with pores of 8 μm. These inserts were placed into 24-well plates. The lower chamber was filled with 800 µL of corresponding culture medium containing 10% FCS. In the case of treatment, cells were incubated in the presence or absence of effectors. Cells were allowed to migrate for 24 h. Migrated cells were fixed and stained with crystal violet. Images were taken and analyzed using image J. Data are expressed as means ± SEM of three independent experiments.

### 2.13. Quantitative Real-Time PCR

Total RNA was extracted from cultured cells using the Qiagen Rneasy Mini kits. Complementary DNA was synthesized using a standard reverse transcription method (qScript cDNA SuperMix, Quanta Biosciences). qPCR reactions were performed using the SYBR Green PCR Master Mix (Thermo Fisher Scientific). The experiments were performed according to the manufacturer’s instructions using QuantStudioTM 3 Thermo Fisher Scientific Real-Time PCR system. The comparative CT method was used to determine relative gene expression levels for each target gene and 18S was used as an internal control for normalization (18S was the most stable gene among 4 reference genes tested). The sequences of the primers used for qPCR are available upon request.

### 2.14. Statistical Analysis

PFS and OS were evaluated using Kaplan–Meier curves using the Cox regression method. Survival statistical analyses were performed using IBM SPSS Statistics 21 (Chicago, IL, USA). A significant *p*-value was <0.05.

IC50 values were calculated from dose–response curves using GraphPad Prism software v6 (GraphPad Software, La Jolla, CA, USA). All data are expressed as means ± SEM of at least three independent experiments. Statistical significance was assessed by the Student’s *t*-test using GraphPad Prism software (* *p* < 0.05, ** *p* < 0.01, *** *p* < 0.001), and combination index (CI) was calculated using the CompuSyn program (Version 1.0, Chou and Martin) (CI < 1, =1, and >1 represent synergistic, additive, and antagonist effects, respectively).

## 3. Results

### 3.1. PHBs Are Associated with Short Patient Survival and Are Highly Expressed in Melanoma Lines Irrespective of BRAF/NRAS Mutational Status

Previous studies reported that PHBs are associated with a poor prognosis in several cancers such as gallbladder and neuroblastoma [7,10]. To determine the implication of PHBs in melanoma progression, PHB1 and PHB2 mRNA expression was evaluated by real-time PCR in the skin and lymph node metastases of 37 patients. After optimizing the cutoff value to define low vs. high PHB groups, the Kaplan–Meier curves and Cox regression analyses demonstrated that the PHB2 mRNA expression was significantly associated with the PFS and OS in the melanoma patient cohort (Figure 1A). Indeed, a high level of PHB2 correlated with shorter PFS (*p* = 0.014) as well as shorter OS (*p* = 0.01). Of note, although not significant, a trend can be observed between PHB1 and PFS (*p* = 0.067). These data clearly indicated the importance of PHBs from a clinical point of view.

In parallel, we evaluated PHB1 and PHB2 protein expressions in three groups of seven melanoma lines each representing the three major molecular subtypes (^WT^BRAF/^WT^NRAS, ^MUT^BRAF, and ^MUT^NRAS). Our data indicated that PHBs are highly expressed in the majority of these lines (Figure 1B,C).

### 3.2. PHB Ligands Inhibit Cell Proliferation in a Large Panel of Melanoma Cells

We studied the effect of PHB ligands MEL 9, 41, and 56 (singled out in the first study) and JI130 on cell proliferation in four different melanoma lines (Appendix A). Considering their low IC50, we selected MEL56 and JI130 for the rest of the study. We found that MEL56 and JI130 inhibit cell proliferation with IC50s ranging from 0.08 to 0.25 µM for JI130 and from 4.7 to 15.5 µM for MEL56 in a large panel of representative melanoma lines comprising three with ^WT^BRAF/^WT^NRAS (HBL, LND1, and MM162), four with BRAF mutations (MM074, MM164, MM029, and MM032) and two with NRAS mutations (MM161 and MM165). The panel included lines with intrinsic resistance to MAPKi as well (Table 1).

### 3.3. PHB Ligands Inhibit Cell Growth and Cell Viability of 3D Melanoma Spheroids

Three-dimensional cultures are clinically relevant models that display structural similarities with human tumors and can reflect more realistically the response to treatment. In this study, we exposed 3D tumor spheroids to PHB ligands. The total spheroid area was evaluated to assess changes in the size and destruction of architecture following treatment. Both JI130 and MEL56 significantly inhibited the growth of melanoma spheroids after 3 days of treatment compared to the control group, as well as a loss of spheroid integrity (Figure 2A,B). They both significantly reduced cell viability by at least 50% (Figure 2C).

### 3.4. PHB Ligands Induce Cell Apoptosis via the Loss of Mitochondrial Potential (MMP) and Caspase Activation

Mitochondrial membrane PHBs have an anti-apoptotic role. Therefore, we tested the effect of PHB ligands on apoptosis (Annexin V staining), the mitochondrial membrane potential (MMP), and caspase activity (Figure 3). Both PHB ligands JI130 and MEL56 induce dose-dependent apoptosis in a panel of representative melanoma lines harboring wild-type BRAF and NRAS (HBL, LND1, and MM162), BRAF mutation (MM074 and MM029) or NRAS mutation (MM165) (Figure 3A). The percentage of apoptotic cells reached 42–62% with 1 µM JI130, while it was 30–48% with 10 µM for MEL56.

Furthermore, we showed that both MEL56 and JI130 induce apoptosis via the loss of mitochondrial membrane potential (MMP), which declined from 93% to 30% along with increasing concentrations of JI130 (0.1 to 1 µM), while it declined from 93 to 52% with 1 to 10 µM for MEL56 (Figure 3B).

The induction of apoptosis was further supported by robust caspase activation. Caspase 3/7 activity was significantly upregulated following treatment with increasing concentrations of JI130 and MEL56 (Figure 3C). The rate of this activity increased from 5% to 55% with 0.1 to 1 µM JI130. Of note, caspase 3/7 activity remained low in the MM165 line compared to all others. MEL56 caused caspase 3/7 activity to increase from 5 to 42% (Figure 3C).

Altogether, these results support that both JI130 and MEL56 induce cell apoptosis via the loss of mitochondrial potential (MMP) and the caspase activation in melanoma lines irrespective of their mutational status.

### 3.5. PHB Ligands Inhibit PHB Expression, the Two Main Survival Pathways MAPK and PI3K/AKT, and Promote p53 Expression in Melanoma Cells

We investigated the effect of PHB ligands on the main survival pathways in melanoma in three representative melanoma lines with different mutational status and observed that JI130 (Figure 4A,B) and MEL56 (Figure 4C,D) inhibit PHBs and the HES1 expression that express the latter.

In accordance with the finding that PHBs activate the MAPK pathway through the activation of CRAF, we show that PHB ligands (JI130 and MEL56) can inhibit pCRAF, and hence, the subsequent phosphorylation of ERK (Figure 4). Noteworthy, this effect was more pronounced in ^WT^BRAF melanoma cells compared to BRAF mutant cells. Indeed, unlike BRAF mutant cells, ^WT^BRAF cells are more dependent on CRAF to activate ERK.

Interestingly, we also show that these ligands inhibit the AKT pathway and promote the p53 expression (Figure 4).

Collectively, these results show that PHB ligands can inhibit both main pathways in melanoma MAPK (CRAF-ERK axis) and AKT, and can reactivate p53.

### 3.6. Autophagy Inhibition Enhances PHB Ligand Antitumor Efficacy in Melanoma Cells

We previously showed that PHB ligands can promote the expression of LC3-II in melanoma cells [25]. Herein, we investigate PHB ligand-induced autophagy and its impact on cell death. First, we confirmed that JI130 and MEL56 increase LC3-II expression in two different melanoma lines (^WTBRAF^HBL and ^MUTBRAF^MM074) (Figure 5A). Then, we assessed autophagy vacuoles and apoptosis after 24 h and 48 h following exposure to JI130 or MEL56. We observed a very significant induction of autophagy 24 h following treatment (a 2–4-fold increase in the fluorescence signal) (Figure 5B) that completely faded away after 48 (Figure 5B). This was the time at which apoptosis reached a maximum level with 41% and 47% of apoptotic cells for JI130 (0.5 µM) and MEL56 (10 µM), respectively (Figure 5C).

Several studies showed that autophagy [28,29] may be a protective mechanism in tumor cells by allowing for them to survive under stressful conditions induced by targeted therapies, and that autophagy inhibition can enhance the anti-tumor efficacy of such drugs. Accordingly, we further investigated cell growth and apoptosis when melanoma cells were challenged with JI130 or MEL56 alone or in combination with the autophagy inhibitor chloroquine (CQ). We found a dramatic reduction in cell proliferation with the combination of CQ (25 μM) with either JI130 (0.01–1 μM) or MEL56 (1–10 μM) as compared to each effector alone (Figure 5D). The latter effect was synergistic as combination index (CI) values ranged from 0.3 to 0.9 (Appendix A). Moreover, such combination increased apoptosis (an increase in the number of apoptotic cells by 1.7- to 3.5-fold for JI30 and by 1.5- to 3.1-fold for MEL56) (Figure 5E).

Altogether, these data show that the inhibition of autophagy highly sensitizes melanoma cells to the proapoptotic effect of PHB ligands by moderating the protective role of autophagy in these conditions.

### 3.7. PHB Ligands Inhibit the Invasive Phenotype in Melanoma Cells

PHBs are known to contribute to EMT-like phenotypes in cancer cells; thus, we assessed the ability of PHB ligands to inhibit the invasive phenotype in melanoma. First, we showed that JI130 and MEL56 inhibit cell migration in two invasive melanoma cell lines (MM029 and MM165) (Figure 6A) and downregulate the mRNA expression of the main EMT/invasion markers such as AXL, EGFR, MET, ZEB1, WNT5A, TGFβ, SNAI1, TWIST and MMPs (Figure 6B). This was also confirmed at the protein level by Western blot, as we noticed an inhibition of the important markers of the invasive phenotype (AXL, ZEB1, and MMP9) in a concentration-dependent manner (Figure 6C).

### 3.8. PHB Ligands Reverse the Acquired Resistance to BRAFi/MEKi Associated with an Up-Regulation of PHBs in BRAF Mutant Melanoma

The development of acquired resistance is a major challenge of MAPK-targeted therapy in melanoma. Here, we tested whether PHBs are associated with BRAFi/MEKi-acquired resistance. First, we established BRAFi/MEKi-resistant cells (MM074-R-Dabrafenib/Trametinib). These cells showed a dramatic increase in the resistance to this combination, illustrated by up to a 4000-fold increase and 600-fold increase in IC50 values of BRAFi and MEKi, respectively (Figure 7A), together with an upregulation of PHBs (Figure 7B). Importantly, we confirmed that PHB mRNA levels are also up-regulated in 19 patient samples following relapse on BRAFi/MEKi compared to matched samples obtained prior to treatment (dataset RNAseq-65185) (Figure 7C).

Moreover, we assessed cell viability and apoptosis in MM074-R (BRAFi/MEKi). As expected, BRAFi (Dabrafenib), MEKi (Trametinib), and their combination have no effect in the resistant cells, while JI130 and MEL56 alone significantly inhibited resistant cell growth. The latter effect was potentiated when PHB ligands were combined with BRAF and MEK inhibitors (Dabrafenib/Trametinib: D/T) (D+T: 94% vs. (D+T+JI130): 53% and 36% or vs. (D+T+MEL56): 72% and 31% obtained with 0.5 µM and 1 µM of JI130 or 1 and 10 of MEL56, respectively) (Figure 7D). Furthermore, these PHB ligands synergized with MAPKi to induce cell apoptosis (38% to 54% increase in apoptotic cells with JI130 and MEL56) (Figure 7E). Finally, Western blotting analysis showed that the addition of PHB ligands to BRAFi/MEKi downregulates the expression of PHBs and HES1, as well as the phosphorylation of CRAF, ERK, and AKT. Interestingly, this combination promotes the upregulation of p53 (Figure 7F).

Taken together, our results demonstrate that PHB ligands can overcome acquired resistance to combined BRAF/MEK inhibitors in BRAF-mutated melanoma.

### 3.9. PHB Ligands Combined with MAPK Inhibitors Synergize to Inhibit Cell Proliferation and Induce Cell Apoptosis in Melanoma Cells with Different Molecular Subtypes

We investigated the effect of JI130 and MEL56 in combination with MAPK inhibitors on cell proliferation in the main molecular subtypes of melanoma: ^WT^BRAF/^WT^NRAS, ^MUT^BRAF and ^MUT^NRAS groups (Figure 8A).

We observed that JI130 and MEL56 enhance the sensitivity to TKI (Sunitinib) in two sensitive ^WT^BRAF/^WT^NRAS lines (HBL and LND1) and in a resistant one (MM162) (Figure 8A). The IC50 of the combination (JI130 or MEL56 with Sunitinib) was 2–9-fold lower compared to Sunitinib alone. In addition, we showed that the combination of these PHB ligands with the BRAF inhibitor Dabrafenib overcame innate resistance in a BRAF mutant melanoma line (MM029), while it was less pronounced in a BRAFi-sensitive line (MM074) (Figure 8A). The IC50 of the combination (JI130 or MEL56 with dabrafenib) was 2–13-fold lower compared to Dabrafenib alone. Moreover, we found that these PHB ligands increase MEKi (Trametinib) sensitivity in an NRAS mutant melanoma line (MM165) (Figure 8A). The IC50 of the combination (JI130 or MEL56 with Trametinib) was 3.2–5-fold lower compared to Trametinib alone.

The combination of PHB ligands and MAPK inhibitors (Sunitinib, Dabrafenib, or Trametinib) displayed a synergistic inhibitory effect in almost all melanoma lines tested (CI values < 1) (Figure 8B).

Moreover, the combination of PHB ligands with MAPKi had significant effects on apoptosis induction in the three main melanoma subtypes (Figure 8C).

Altogether, these data provide a rationale to target PHBs in combination with MAPKi as a new therapeutic strategy in melanoma.

## 4. Discussion

Mitochondria play a pivotal role in melanoma progression and metastasis dissemination. They provide bioenergetic flexibility needed in changing TME and under therapy. This metabolic plasticity confers resistance to targeted therapies in melanoma. Prohibitins (PHBs) are evolutionarily conserved proteins overexpressed in several cancers and implicated in cancer development [3,20]. They are largely found in the mitochondrial membrane and control mitochondrial integrity, metabolism, and apoptosis. All of the above suggest that there is a rationale to develop strategies to target the mitochondria in melanoma using PHB ligands. Furthermore, PHBs have been reported to regulate cell survival through different pathways. For instance, PHBs are required for CRAF-mediated MAPK/ERK1/2 activation and they can stabilize RACK1, which induces activation of the AKT [9,12]. Thus, these two main pathways in melanoma can be targeted at once.

In the present study, we investigated PHB roles in further detail using specific ligands by putting together and analyzing various effects on apoptosis, autophagy, MAPK pathways, invasive phenotype, and resistance to targeted drugs in a representative panel of melanoma cell lines harboring different mutational statuses.

First, we found that PHBs are highly expressed in melanoma and that high PHB2 significantly correlates with poor patient outcomes, supporting the importance of PHBs in melanoma progression. In accordance, it was demonstrated that high PHB2 expression was observed in NSCLC patients with an advanced clinical stage (stages III/IV) compared to those with an early clinical stage (stages I/II) [9].

Then, we evaluated the effect of novel PHB ligands/inhibitors in melanoma. Such molecules are melanogenin analogs (MEL56) identified in our previous screening study, and JI130 is a new compound that has been found to inhibit HES1 via the interaction with PHB2. Hence, we showed that targeting PHBs using these novel PHB inhibitors can inhibit cell proliferation in a panel of representative cell lines including those with intrinsic resistance to MAPK-targeted agents. Moreover, we investigated the cell survival in 3D cell culture models that may more accurately mimic in vivo tumors and thus provide more reliable data for subsequent drug testing. We validated that PHB ligands are able to potently inhibit tumor growth in 3D melanoma cultures.

Furthermore, we showed that these PHB ligands can induce cell apoptosis via the loss of mitochondrial potential (MMP) and caspase activation in many melanoma cells irrespective of BRAF/NRAS mutation status. This observation is in line with several studies demonstrating the anti-apoptotic role of PHBs through mitochondria activity alterations. Indeed, the accumulation of PHBs within mitochondria in cancer cells leads to resistance to apoptosis. Also, PHBs stabilize anti-apoptotic factors such as HAX1, and a decreased expression of PHB2 is associated with a loss of mitochondrial integrity and the activation of caspases [30,31]. Therefore, apoptosis can be considered to be the most important mechanism underlying the anti-tumor activity of PHB ligands. However, it is not the sole determinant of cell fate. Indeed, autophagy has also a crucial role in cell death decisions and can protect tumor cells by preventing them from undergoing apoptosis. In our previous study, we showed that PHB ligands were able to regulate cell differentiation via LC3 activation, a hallmark of autophagy induction. Herein, we show that autophagy inhibition enhances the antitumor efficacy of PHB ligands in melanoma cells. Autophagy is a phenomenon involved in cell survival as well as cell death, depending on both cell context and stress levels [32,33]. Indeed, it allows for cancer cells to survive under stressful conditions induced by therapeutic agents, but prolonged autophagy can lead to cell death through apoptosis activation [34,35]. Here, we found that inhibition of autophagy increases the sensitivity to PHB ligands in melanoma cells, indicating a protective role of autophagy in PHB-induced cell death.

Moreover, our findings reveal that PHB ligands inhibit the EMT-like invasive phenotype in melanoma cells. Accordingly, PHBs promote a dedifferentiated phenotype in neuroblastoma [7]. In addition, PHBs support migration, invasion, and EMT in breast, pancreatic, and lung cancer models. PHBs also promote metastasis development in xenograft models [7,9,11,12,15]. Therefore, our data support the fact that PHBs are key contributors to metastasis and EMT phenotypes in cancer. It is noteworthy that PHBs are implicated in cancer cell proliferation, survival, and invasion, which appear to be in a cellular context-dependent manner.

In addition, we investigated the role of these PHB ligands on the main pathways in melanoma, given that several studies emphasize the role of PHBs in key melanoma signaling pathways, providing a new avenue to target these pathways concurrently. Indeed, we found that JI130 and MEL56 inhibit the two main survival pathways MAPK (CRAF-ERK axis) and PI3K/AKT, and promote p53 expression in melanoma. Consistently, it was demonstrated that Fluorizoline, a cytotoxic drug which binds to PHB1/2, can also induce p21 expression a main target of p53 [23].

All the above results support a possible potentiating role for PHB ligands when combined with MAPK inhibitors to overcome both innate and acquired resistances. The acquired resistance to MAPKi is considered as the main obstacle in melanoma treatment. Indeed, most patients after an initial impressive response to MAPKi develop acquired resistance and relapse. In our study, we found that the acquired resistance to the combination of BRAFi/MEKi is associated with an upregulation of PHBs, and that PHB ligands can overcome this resistance. Interestingly, we also showed that the combination of PHB ligands with MAPK inhibitors (TKi, BRAFi, and MEKi) display a synergistic effect in terms of growth inhibition and apoptosis induction in melanoma cells with different molecular subtypes including ^WT^BRAF/^WT^NRAS, BRAF mutant (including innate resistance line), and NRAS mutant melanoma.

In conclusion, innate and acquired resistance to MAPK-targeted agents in BRAF-mutated melanomas and poor therapeutic options in the other subtypes make the identification of new targets highly important. Our study shows that PHBs represent a key therapeutic target in melanoma given their association with poor patient survival and relapse, as well as their role in multiple processes including cell proliferation, migration, and apoptosis. We also propose novel therapeutic strategies targeting PHBs in combination with current targeted agents, including MAPKi, which may be very promising in melanoma.

## Figures and Tables

**Figure 1 cells-12-01855-f001:**
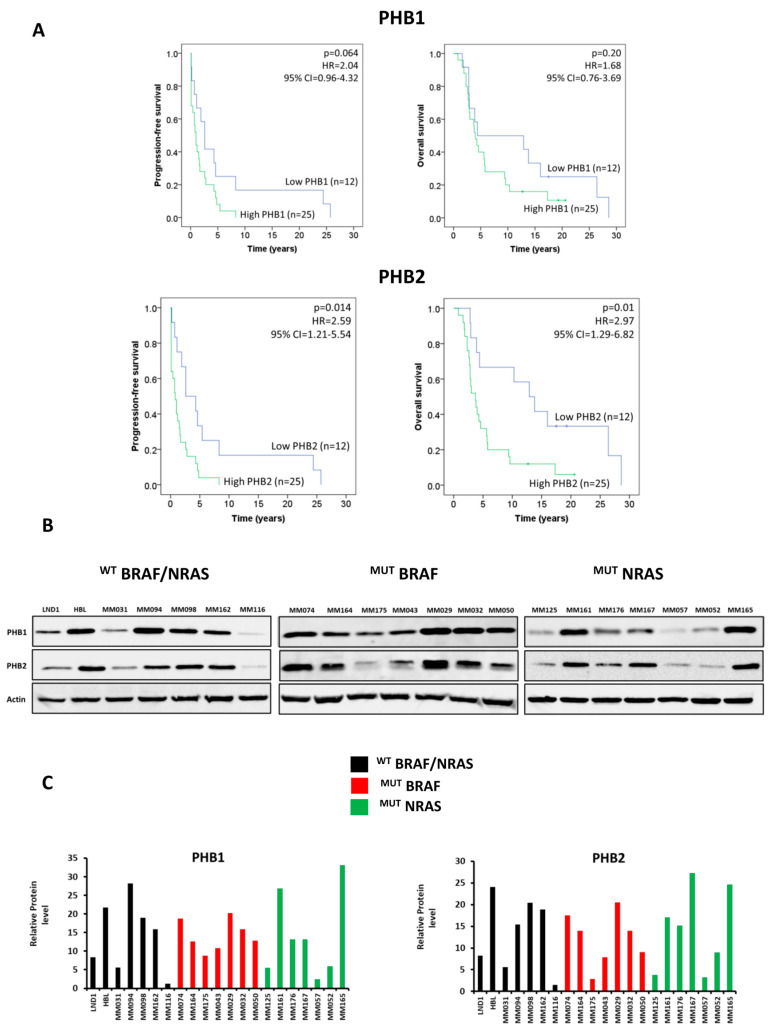
PHBs are highly expressed in melanoma irrespective of BRAF/NRAS status and are associated with worse survival. (**A**) Kaplan–Meier curves and Cox regression evaluation for overall survival and progression-free survival of melanoma patients with either high or low expressions of PHB1 and PHB2. (**B**,**C**) Evaluation of the protein expression of PHB1 and PHB2 in 21 melanoma lines: 7 lines with ^WT^BRAF/^WT^NRAS (black), BRAF mutant lines (red), and NRAS mutant lines (green).

**Figure 2 cells-12-01855-f002:**
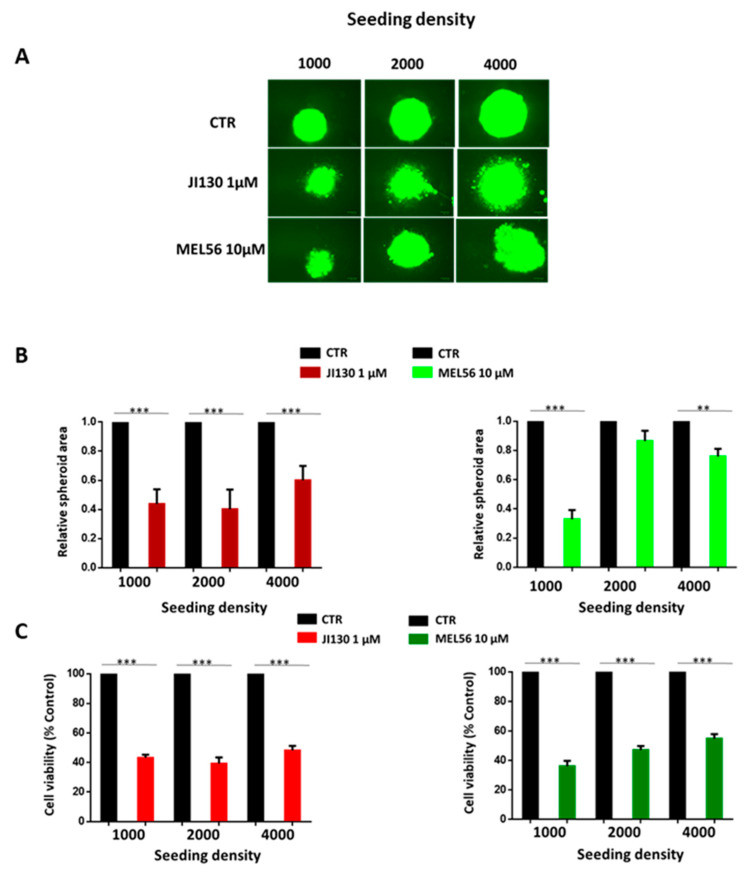
PHB ligands (JI130 and MEL56) inhibit cell growth in 3D melanoma cultures. (**A**) Representative images of HBL melanoma spheroids (transfected with GFP vector). (**B**) Graph represents the relative spheroid size (spheroid area). (**C**) Graph represents the relative cell viability. Data are presented as means ± SEM from three independent experiments (** *p* < 0.01; *** *p* < 0.001, *t*-test).

**Figure 3 cells-12-01855-f003:**
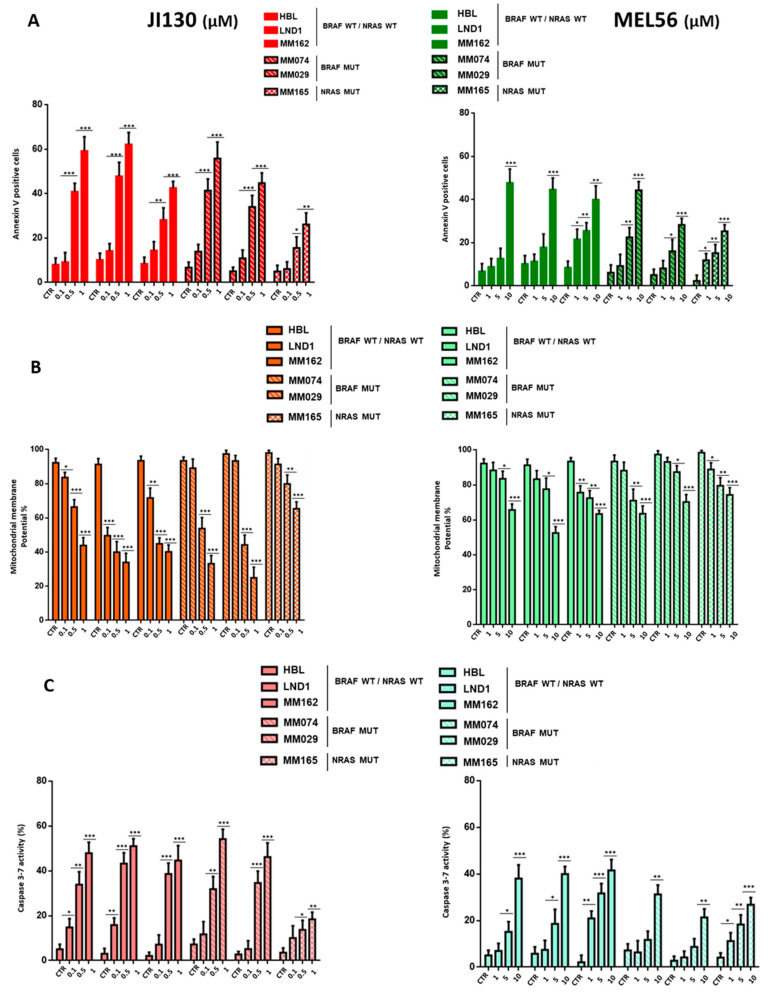
PHB ligands (JI130 and MEL56) induce cell apoptosis in a panel of representative melanoma lines. Effect of JI130 (0.1, 0.5 and 1 µM) and MEL56 (1, 5 and 10 µM) on cell apoptosis (**A**) (annexin-V-positive cells), (**B**) mitochondrial potential (MMP) (fluorescence intensity of MitoProbe DiIC1) and (**C**) caspase-3/7 activity in a panel of representative melanoma lines: ^WT^BRAF/^WT^NRAS lines (HBL, LND1, and MM162), BRAF mutant lines (MM074 and MM029) and NRAS mutant lines (MM165). Data are presented as means ± SEM (*n* = 3) compared to untreated cells (* *p* < 0.05; ** *p* < 0.01; *** *p* < 0.001, *t*-test).

**Figure 4 cells-12-01855-f004:**
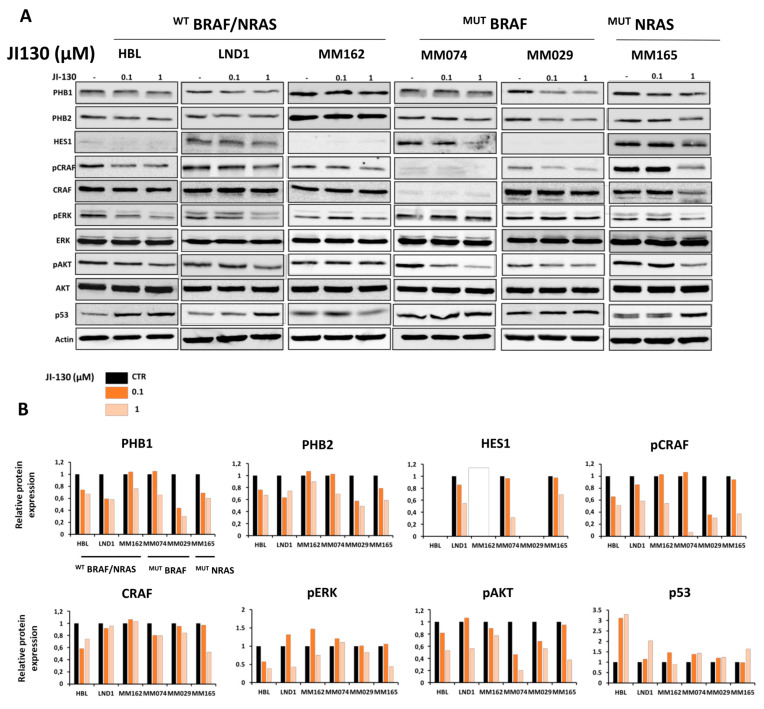
PHB ligands inhibit PHBs expression, the two main survival pathways MAPK and PI3K/AKT, and promote p53 expression in melanoma cells. (**A**,**C**) Western blot analyses of PHBs (PHB1, PHB2), HES1, MAPK, and PI3K/AKT pathway-related proteins and p53 after treatment with the indicated concentrations of JI130 and MEL56 for 24 h in a panel of representative melanoma lines: ^WT^BRAF/^WT^NRAS lines (HBL, LND1, and MM162), BRAF mutant lines (MM074 and MM029) and NRAS mutant lines (MM165). (**B**,**D**) Western blot quantification shows the signal intensities of proteins normalized to β-actin and relative to control.

**Figure 5 cells-12-01855-f005:**
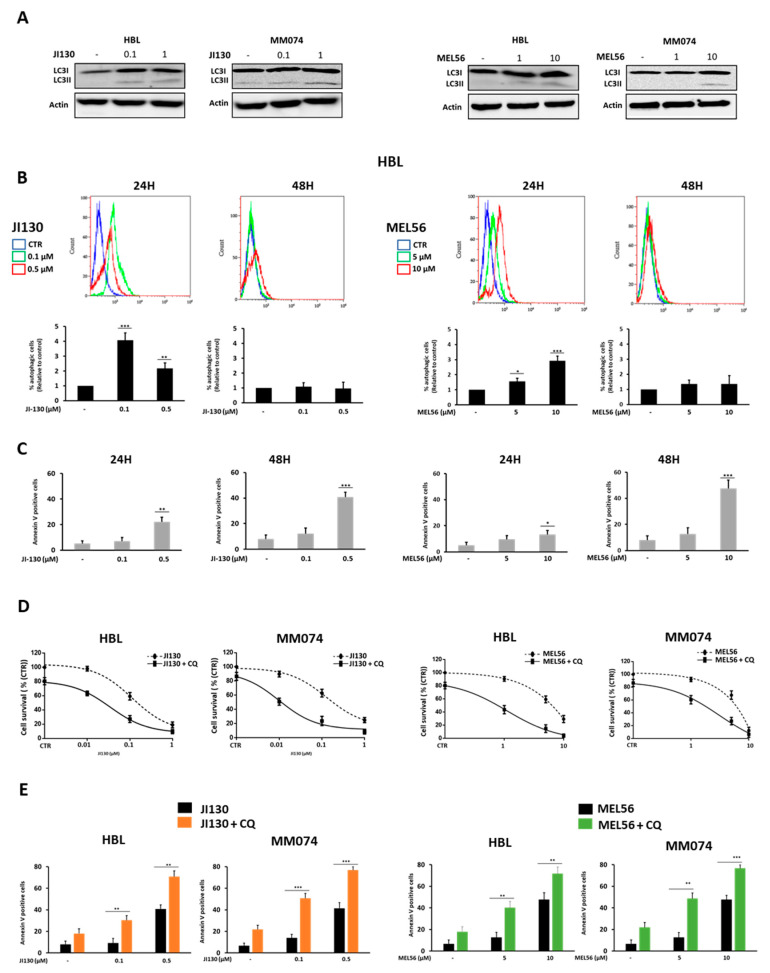
Autophagy inhibition enhances the antitumor efficacy of PHB ligands in melanoma cells. (**A**) Western blot analyses of LC3I and LC3II after treatment with the indicated concentrations of JI130 and MEL56 for 24 h in HBL and MM074 melanoma lines. (**B**) Autophagy vacuole determination. Representative flow cytometry plots illustrating the median fluorescence intensity (MFI) of green fluorescence compared to the control after treatment with PHB ligands (MEL56 and JI130) for 24 and 48 h in HBL lines (top panel). The percentage of autophagic cells was represented relative to the control (bottom panel). (**C**) Effect of PHB ligands JI130 (0.1 and 0.5 µM) and MEL56 (5 and 10 µM) on apoptosis (annexin-V-positive cells) for 24 and 48 h. (**D**,**E**) Effect of the treatment of PHB ligands JI130 and MEL56 with CQ (25 µM) on (**D**) cell growth and (**E**) cell apoptosis. Data are presented as means ± SEM for three independent experiments (* *p* < 0.05; ** *p* < 0.01; *** *p* < 0.001, *t*-test).

**Figure 6 cells-12-01855-f006:**
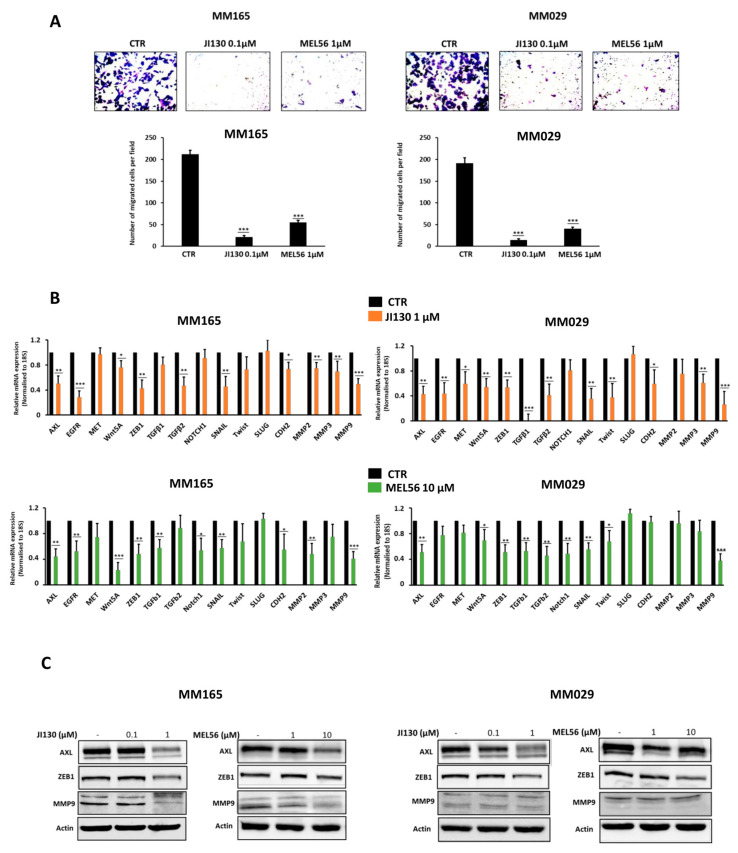
PHB ligands inhibit the invasive phenotype in melanoma cells. (**A**) Effect of JI130 and MEL56 on cell migration activity in two different invasive melanoma cells (MM029 and MM165). The upper panel shows representative regions of the chamber filters with crystal violet-stained cells. The number of migrated cells per field was calculated from three independent experiments. (**B**) Relative mRNA expression by real-time quantitative PCR and (**C**) protein expressions (representative Western blot) of the main invasive markers in MM029 and MM165 melanoma cells following the treatment of PHB ligands (JI130 and MEL56). Data are presented as means ± SEM for three independent experiments (* *p* < 0.05; ** *p* < 0.01; *** *p* < 0.001, *t*-test).

**Figure 7 cells-12-01855-f007:**
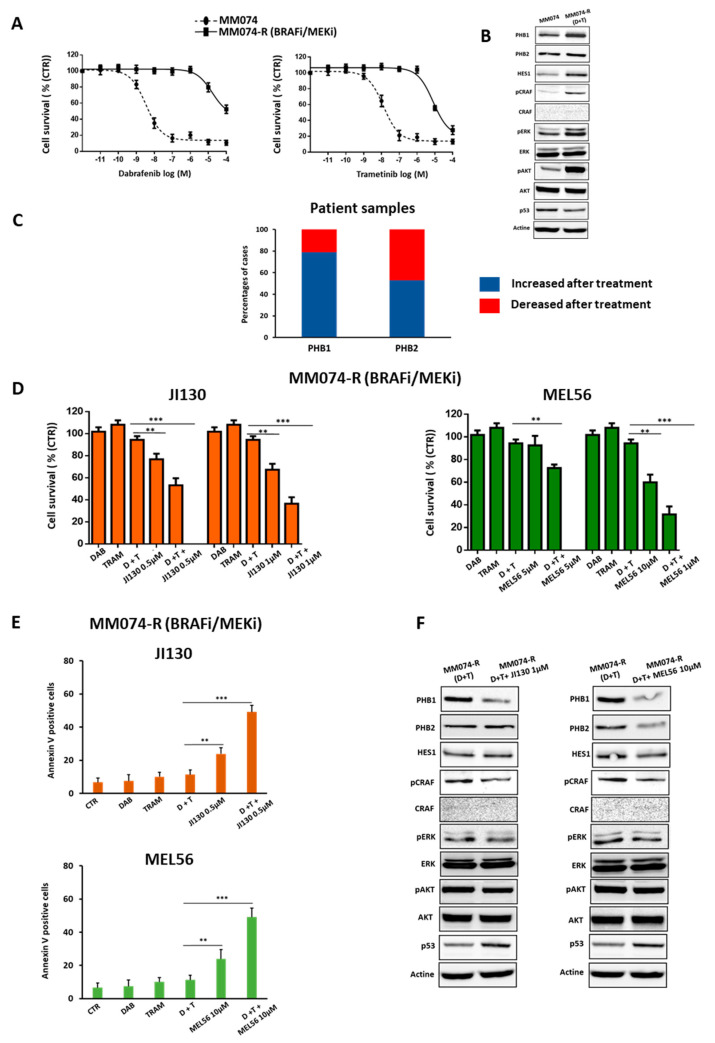
PHB ligands reverse the acquired resistance to BRAFi/MEKi associated with an up-regulation of PHBs in BRAF mutant melanoma. (**A**) Effect of increasing concentrations of BRAFi (Dabrafenib: 10^−11^–10^−4^ M) and MEKi (Trametinib:10^−11^–10^−4^ M) on cell survival in MM074 lines (parental cells) and MM074-R lines (cells with acquired resistance to BRAFi/MEKi). (**B**) Protein expression levels of PHBs (PHB1, PHB2), HES1, MAPK, and PI3K/AKT pathway-related proteins and p53 in MM074 and MM074-R (Dabrafenib/Trametinib: D+T) cells. (**C**) The melanoma dataset (RNAseq-65185) was analyzed for PHB1 and PHB2 mRNA levels pre-/post-BRAFi/MEKi treatment. The number of patient samples: 19. (**D**,**E**) Effect of PHB ligands JI130 (0.5 and 1 µM) or MEL56 (1 and 10 µM) alone or in combination with the combination (Dabrafenib: 1 µM/Trametinib: 0.01 µM) on (**D**) cell survival (crystal violet) and (**E**) cell apoptosis (annexin-V-positive cells) in MM161 parental cells and MM161-R (cells with acquired resistance). (**F**) Protein expression levels as in (B) in MM074-R (Dabrafenib/Trametinib: D+T) cells treated with the combination (D+T) alone or in the presence of PHB ligands JI130 (1 µM) or MEL56 (10 µM). Data are presented as means ± SEM from three independent experiments ( ** *p* < 0.01; *** *p* < 0.001, *t*-test).

**Figure 8 cells-12-01855-f008:**
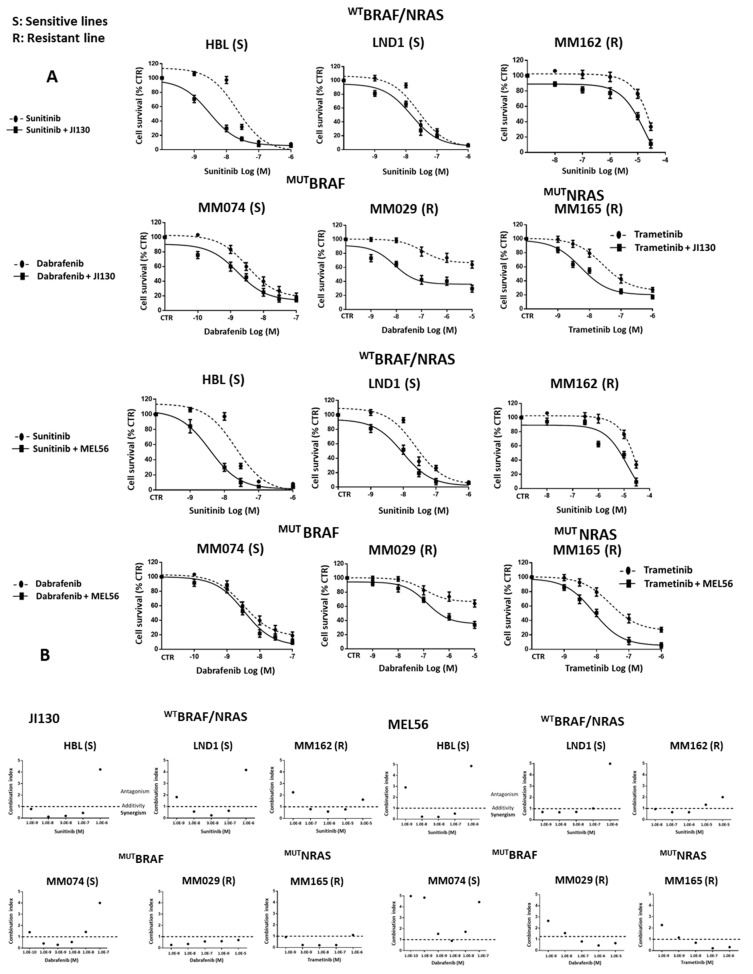
PHB ligands combined with MAPK inhibitors synergize to inhibit cell proliferation and induce cell apoptosis in melanoma lines of different molecular subtypes. (**A**) Effect of increasing concentrations of MAPK targeted agents (TKI (Sunitinib), BRAFi (Dabrafenib) and MEKi (Trametinib) for 3D alone or in combination with 0.005 µM JI130 or 5 µM MEL56 on cell proliferation in a panel of representative melanoma lines composed of three lines with ^WT^BRAF/^WT^NRAS (HBL, LND1, and MM162), two lines with BRAF mutations (MM074 and MM029) and one line with NRAS mutations (MM165). Data are expressed as means ± SEM (*n* = 3) compared to untreated cells (CTR). (**B**) Combination index (CI) analysis. CI plots of the combination of PHB ligands JI130) and MEL56 with different concentrations of MAPK targeted agents in all the lines tested above. CI < 1, CI = 1, and CI > 1 indicate synergism, additive effect, and antagonism, respectively. (**C**) Effect of the combination of PHB ligands JI130 (0.3 µM) and MEL56 (5 µM) with the indicated concentration of MAPKi (TKI (Sunitinib), BRAFi (Dabrafenib) and MEKi (Trametinib) on cell apoptosis (annexin-V-positive cells) in the same panel of lines tested. Data are presented as means ± SEM from three independent experiments (* *p* < 0.05; ** *p* < 0.01; *** *p* < 0.001, *t*-test).

**Table 1 cells-12-01855-t001:** IC50 of the main candidates (MEL56 and JI130) in a large panel of melanoma lines with different mutations. The panel comprised the three major molecular subtypes: ^WT^BRAF/^WT^NRAS, BRAF mutant, and NRAS mutant lines, including those with intrinsic resistance to MAPKi.

	Cell Lines	Resistance to MAPKi	IC50 (µM) JI130	IC50 (µM) MEL56
Wild-type BRAF/NRAS cell lines	HBL	sensitive	0.10	6.1
	LND1	sensitive	0.08	4.7
	MM162	NA	0.16	8.4
BRAF mutant cell lines	MM074	sensitive	0.09	5.0
	MM164	intermediate	0.11	9.3
	MM029	intrinsic	0.17	12.6
	MM032	intrinsic	0.20	10.2
NRAS mutant cell lines	MM161	sensitive	0.25	15.5
	MM165	Low sensitivity	0.18	13.0

## Data Availability

Not applicable.

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
