# Peer review of "Targeting Prohibitins to Inhibit Melanoma Growth and Overcome Resistance to Targeted Therapies"

_cells, 2023, doi:10.3390/cells12141855_

Round 1
Reviewer 1 Report
The article addresses the urgent need for effective targeted therapies in metastatic melanoma treatment. It introduces Prohibitins (PHBs) as potential targets and presents novel PHB inhibitors that show promise in inhibiting melanoma cell growth, including resistant subtypes. The inhibitors demonstrate modulation of key signaling pathways, induction of apoptosis, and potential synergy with MAPKi. The study's comprehensive analysis and possible implications for overcoming drug resistance make it a valuable contribution to melanoma research.
Below you will find some comments that I would like to be addressed by the authors.
1. Figure 1A section PHB2 Overall Survival. The time axe states "hours." It should be years.
2. The authors stated that the overall survival ranges from 0.8 to 28.6 years. Is this range correct, as this is not agreed with the general statistics reported in public databases? If that is the case, let's say that the earliest collected samples in 1998 give an OS to 2026. Please clarify this.
3. Please re-organize Figure 2 as sections D, E, and F are mentioned before the rest of the figure in the text.
4. Lines 289-291. The author concludes that all melanoma cell lines are sensitive to the PHB inhibitors JI130 and MEL56, irrespective of their mutation status. However, cell line MM165, an NRAS mutant, is less responsive than the other cell lines. Please consider this observation and discuss it.
5. Figure 3. The absence of expression of HES1 is an intrinsic effect of the cell lines HBL, MM162, and MM029, as even in the absence of the drug, the protein is not expressed.
6. The western-blot image for HES1 in cells MM162 shows a faint band when cells are treated with JI130 1uM. However, the bar graphs in section B show the total absence of expression on that cell line. Please clarify.
7. Lines 325-326. "These results show that PHB ligands can inhibit both main pathways in melanoma MAPK (CRAF-ERK axis), and AKT, and reactivate p53." This effect is not real in the mutant BRAF cell lines. Please discuss this deeper.
8. Figure 4D. Did the authors test the effect of chloroquine alone? If the test were not performed, I would suggest doing it to demonstrate the synergistic effect of the drugs.
9. Figure 5. Once again, the effects of the tested drugs are not reflected in the mutant BRAF cell line MM029. Is the mutation in BRAF involved in the mechanism of action of the drugs?
Author Response
We would like to thank the editor and the reviewers for their time and their very thoughtful comments. These comments and corrections were all valuable and very helpful to improve our manuscript. The revised manuscript was submitted with tracked changes and the main corrections in the paper and the answers to the Editor’s and reviewer’s comments are as flowing:
- Figure 1A section PHB2 Overall Survival. The time axe states "hours." It should be years:
Thanks for pointing this out. The error has been corrected.
- The authors stated that the overall survival ranges from 0.8 to 28.6 years. Is this range correct, as this is not agreed with the general statistics reported in public databases? If that is the case, let's say that the earliest collected samples in 1998 give an OS to 2026. Please clarify this.
We confirm the accuracy of this range within our cohort. Two patients in our study exhibited long overall survival (OS) durations of 26.4 and 28.6 years. In the publicly available cBioPortal database, specifically in the Melanomas dataset (TCGA, Cell 2015), we examined samples with mutation data (346 samples/344 patients) for PHB1 and PHB2. Within this dataset, we identified patients with overall survival exceeding 350 months (>29.2 years).
Regarding the time of sample collection, we collected metastatic tissues between 1998 and 2009, while the calculation of OS was based on the time of diagnosis of the primary lesions. For instance, in our study, the patient with an OS of 28.6 years had initially developed a primary melanoma in 1979, passed away in 2007, and the metastatic tissue used for analysis was collected in 2004. To provide further clarification in the text, we explicitly mentioned that "metastatic tissue samples" were utilized for analysis (line 93).
- Please re-organize Figure 2 as sections D, E, and F are mentioned before the rest of the figure in the text.
We arrange the orders of the figures.
- Lines 289-291. The author concludes that all melanoma cell lines are sensitive to the PHB inhibitors JI130 and MEL56, irrespective of their mutation status. However, cell line MM165, an NRAS mutant, is less responsive than the other cell lines. Please consider this observation and discuss it.
We agree. The MM165 cell line is less responsive than the other lines, although we still observe an effect of PHB inhibitors. This less pronounced effect may due the fact that MM165 is an invasive cell line and the effect of PHBi are more observed on the invasive phenotype. Thus, PHBi can exert both effects on cell survival and cell migration depending on the cellular context. Indeed, PHBs are implicated in cancer cell proliferation, survival, and invasion, which appear to be cellular context-dependent manner. This will be discussed in the manuscript and the modifications will be highlighted in the manuscript (Track changes).
- Figure 3. The absence of expression of HES1 is an intrinsic effect of the cell lines HBL, MM162, and MM029, as even in the absence of the drug, the protein is not expressed.
- The western-blot image for HES1 in cells MM162 shows a faint band when cells are treated with JI130 1uM. However, the bar graphs in section B show the total absence of expression on that cell line. Please clarify
Indeed, as the reviewer mentions HES1 is not expressed in all melanoma lines and PHBi can inhibit the expression of HES1 in melanoma line that express the protein. A sentence has been added in the manuscript (Track changes).
We thank the reviewer for point out this issue. We adjusted the bar graphs in section B. The bar graph displays the relative expression of proteins relative to control. As the reviewer mentions HES1 is not expressed in HBL, MM162, and MM029, as even in the absence of the drug, compared to LND1, MM074 and MM165 that display high expression of HES1.
- Lines 325-326. "These results show that PHB ligands can inhibit both main pathways in melanoma MAPK (CRAF-ERK axis), and AKT, and reactivate p53." This effect is not real in the mutant BRAF cell lines. Please discuss this deeper.
We completely agree with the reviewer. Indeed, PHBs activate the MAPK pathway through the activation of CRAF-ERK axis. Thus, this effect of PHB ligands was more pronounced in WTBRAF melanoma cells compared to BRAF mutant cells. Indeed, unlike BRAF mutant cells, WTBRAF cells including (WTBRAF/WT NRAS and NRAS mutant cells) are uniquely dependent upon CRAF rather than BRAF for activation of downstream MEK/ERK signaling. Despite the fact, that we did not observe effect on pERK, we observed an inhibition of the phosphorylation of pAKT. This will be discussed in the manuscript and the modifications will be highlighted in the manuscript (Track changes).
- Did the authors test the effect of chloroquine alone? If the test were not performed, I would suggest doing it to demonstrate the synergistic effect of the drugs.
The effect of chloroquine alone was tested and the synergistic effect of the drugs are presented in the supplementary figure 1.
Figure 5. Once again, the effects of the tested drugs are not reflected in the mutant BRAF cell line MM029. Is the mutation in BRAF involved in the mechanism of action of the drugs?
We tested the effect of the drugs in two invasive melanoma lines and we could observe that PHB inhibitors inhibit the invasive phenotype (cell migration, main markers of invasive phenotype) in the two lines tested including MM029.

Reviewer 2 Report
This research article investigates the role of prohibitins (PHBs) in melanoma and explores the potential of PHB ligands as therapeutic agents. The study highlights the limitations of current treatments and the need for effective targeted therapies, particularly for BRAF wild-type melanoma. The study evaluates the efficacy of PHB ligands in inhibiting cell proliferation, inducing apoptosis, suppressing survival pathways, and inhibiting the invasive phenotype of melanoma cells. The results demonstrate the potential of PHB ligands as therapeutic agents for melanoma, especially in combination with MAPK inhibitors. The discussion emphasizes the rationale for targeting PHBs. Overall, the findings support targeting PHBs as a promising strategy to overcome resistance and inhibit tumor progression in melanoma.
However, before publication, several concerns should be addressed regarding the research findings:
1. The research group previously demonstrated that PHB ligands induce apoptosis in various cancer cell lines, including melanoma cells, by inhibiting the AKT survival pathway. The novelty of this study lies in the combination therapy of PHB ligands with MAPK inhibitors. The authors also found that treatment with PHB ligands increased p53 levels in melanoma cells, which is usually wild-type (wt) but mutated in a small percentage of cases. Combining p53-activating agents like nutlin-3a, which inhibits p53-MDM2 interaction, with BRAF and MEK inhibitors could be a promising therapeutic approach for melanoma. It would be interesting to investigate whether there is any synergy between nutlin-3a and PHB analogs in killing melanoma cells.
2. It is important to determine if the drugs used in the combination therapy also have an impact on normal melanocytes. Assessing the potential effects on normal cells is crucial for evaluating the therapeutic selectivity of the treatment.
3. The mechanism by which the ligand degrades PHB1/2 is not clearly explained in the introduction or discussions. It would be helpful to provide more information on this aspect.
4. The results section needs better organization and flow. For example, Figure 2D is presented first and explained, while Figures 2A-C are discussed later. It would be beneficial to rearrange the figures or text to ensure a logical sequence of presentation.
5. In Figure 2E, it is observed that MEL56 relative spheroid area was not significantly inhibited at a seeding density of 2000. It would be valuable to provide an explanation for this finding and discuss any potential reasons for the lack of significant inhibition in this particular condition.
Author Response
We would like to thank the editor and the reviewers for their time and their very thoughtful comments. These comments and corrections were all valuable and very helpful to improve our manuscript. The revised manuscript was submitted with tracked changes and the main corrections in the paper and the answers to the Editor’s and reviewer’s comments are as flowing:
The authors also found that treatment with PHB ligands increased p53 levels in melanoma cells, which is usually wild-type (wt) but mutated in a small percentage of cases. Combining p53-activating agents like nutlin-3a, which inhibits p53-MDM2 interaction, with BRAF and MEK inhibitors could be a promising therapeutic approach for melanoma. It would be interesting to investigate whether there is any synergy between nutlin-3a and PHB analogs in killing melanoma cells
We totally agree with the reviewer; and we would like to express our sincere appreciation for the remarkable and the great idea you shared. Indeed, combining p53-activating agents like nutlin-3a, which inhibits p53-MDM2 interaction, with BRAF and MEK inhibitors could be a promising therapeutic approach for melanoma. Indeed, we already demonstrate that promising effects in our previous studies ( (Krayem et al., 2016, 2019; Najem et al., 2017). Regarding, the effect of combining p53-activating agents with PHB ligands, it will be the main subject of our next article where will investigate this combination in deep.
It is important to determine if the drugs used in the combination therapy also have an impact on normal melanocytes. Assessing the potential effects on normal cells is crucial for evaluating the therapeutic selectivity of the treatment.
We found that the PHB ligands developed don’t show any cytotoxicity in human normal cells (human fibroblast lines) even at higher concentration than those used in our study.
The mechanism by which the ligand degrades PHB1/2 is not clearly explained in the introduction or discussions. It would be helpful to provide more information on this aspect.
In our previous study, we demonstrated that these PHB ligands can bind to PHB and here we found that these PHB ligands can downregulate the cellular levels of PHB protein. In other reports, they identified that natural PHB ligand can lead to the diminution of the amount of mitochondria-associated PHB and can interfere with membrane localization of PHB. This will be discussed in the manuscript (introduction) and the modifications will be highlighted in the manuscript (Track changes).
The results section needs better organization and flow. For example, Figure 2D is presented first and explained, while Figures 2A-C are discussed later. It would be beneficial to rearrange the figures or text to ensure a logical sequence of presentation.
Thank you. We arrange the orders of the figures.
In Figure 2E, it is observed that MEL56 relative spheroid area was not significantly inhibited at a seeding density of 2000. It would be valuable to provide an explanation for this finding and discuss any potential reasons for the lack of significant inhibition in this particular condition.
We agree the relative area was not significantly inhibited at a seeding density of 2000. However, we could notice the disruption of the architecture and the inhibition of cell viability in the spheroids at this seeding density.

Round 2
Reviewer 2 Report
While the novelty of this study may be limited, it holds significant importance for melanoma therapy. Therefore, I suggest publishing the article.